# Restore3D: Breathing Life into Broken Objects with Shape and Texture Restoration

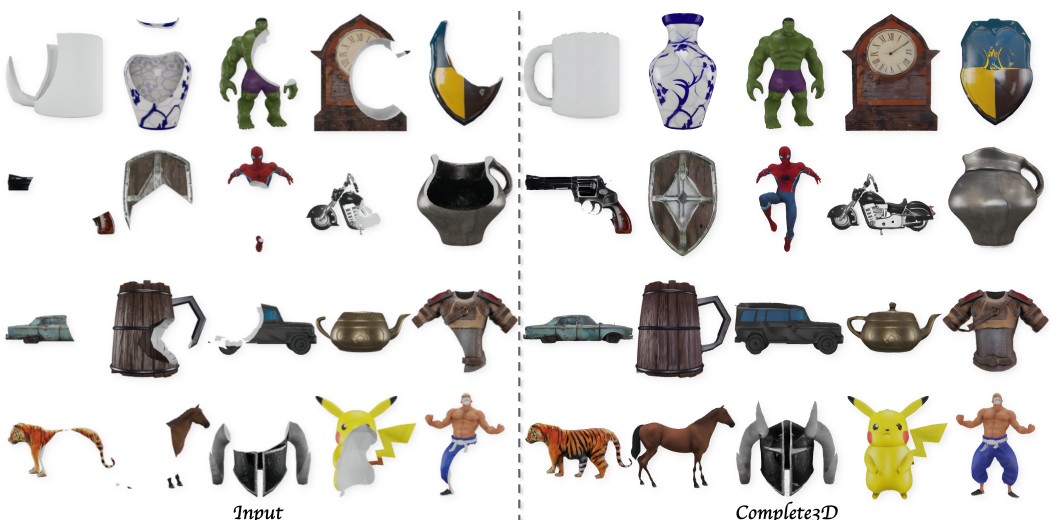

Figure 1: Completion Results. Our **Restore3D** is among the first to simultaneously restore the shape and texture of relatively complex and diverse objects, producing highly plausible and realistic results.

## Abstract

Restoring incomplete or damaged 3D objects is crucial for cultural heritage preservation, occluded object reconstruction, and artistic design. Existing methods primarily focus on geometric completion, often neglecting texture restoration and struggling with relatively complex and diverse objects. We introduce Restore3D, a novel framework that simultaneously restores both the shape and texture of broken objects using multi-view images. To address limited training data, we develop an automated data generation pipeline that synthesizes paired incomplete-complete samples from large-scale 3D datasets. Central to Restore3D is a multi-view model, enhanced by a carefully designed Mask Self-Perceiver module with a Depth-Aware Mask Rectifier. The rectified masks, learned through the self-perceiver, facilitate an image integration and enhancement phase that preserves shape and texture patterns of incomplete objects and mitigates the low-resolution limitations of the base model, yielding high-resolution, semantically coherent, and view-consistent multi-view images. A coarse-to-fine reconstruction strategy is then employed to recover detailed textured 3D meshes from refined multi-view images. Comprehensive experiments show that Restore3D produces visually and geometrically faithful 3D textured meshes, outperforming existing methods and paving the way for more robust 3D object restoration. Project page: https://nip-ss.github.io/NIPS-anonymous/.

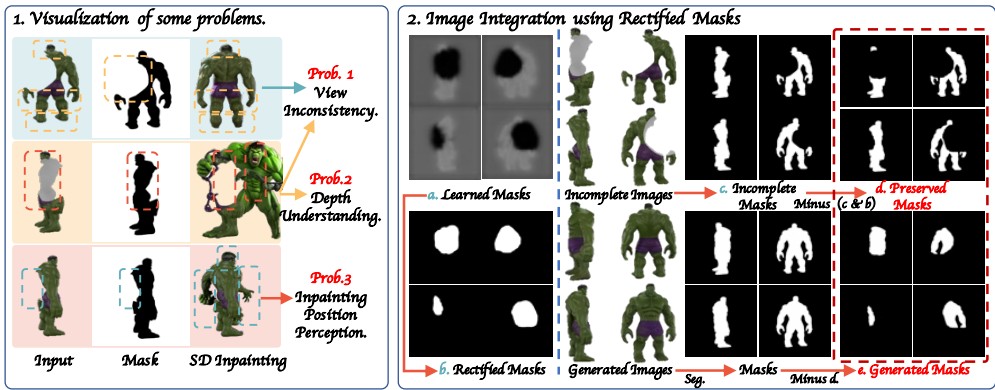

Figure 2: **The importance of masks.** In single-view inpainting, user-provided masks define the regions requiring inpainting. However, in a multi-view context, manually creating consistent masks across all views is impractical. Directly inverting object masks to serve as inpainting masks inevitably causes issues (see Prob. 1 & 3). Moreover, manually adjusting masks based on depth information (see Prob. 2) is labor-intensive and time-consuming. As shown in the right figure (a), our mask self-perceiver can automatically indicate the regions that need to be completed. By leveraging both preserved and generated masks (d & e), our approach retains the incomplete object's patterns, ensuring accurate and consistent multi-view inpainting. These masks are also used for the image enhancement stage to yield high-resolution restored images (see Fig. 4).

# 1 Introduction

Recent advances in 3D generation and reconstruction techniques [12, 45, 30, 29, 69, 31, 56] have demonstrated impressive capabilities, paving the way for innovative applications across diverse fields. Despite these strides, a significant gap remains in the comprehensive restoration of both shape and texture for broken or incomplete 3D objects. This challenge is particularly relevant for some applications such as cultural heritage preservation, occluded objects reconstruction, and artistic creation, where high-fidelity restoration/completion is crucial.

In this study, we aim to develop a robust framework that can simultaneously restore the shape and texture of incomplete 3D objects while handling complex and diverse data types. Key challenges in achieving this goal include: *i) Data Collection*. Existing 3D datasets [6, 16, 48] focus primarily on shape completion, often neglecting the equally critical aspect of texture restoration. Furthermore, these datasets typically contain simple objects. Creating a diverse, high-quality dataset remains labor-intensive and time-consuming. *ii) Complexity of Object Completion*. Addressing the intricacies of restoring complex and general objects requires a robust framework, as simpler methods often fall short. *iii) Consistency Preservation of Broken Parts*. Incomplete objects may exhibit varying degrees of degradation in shape and texture. Therefore, preserving the integrity of original components, including consistent color, style, and structural coherence, is crucial for realistic restoration.

To address these challenges, we propose several complementary solutions: **i) Synthetic Data Generation**. To overcome the limitations of existing datasets, we propose to synthesize paired broken and complete data. **ii) Leveraging Foundation Models**. Recent advancements in foundation models [23, 52, 50, 43, 28, 71] have demonstrated exceptional generalizability, due to their extensive architectures, large-scale datasets, and adaptability through fine-tuning. We incorporate foundation models to provide prior knowledge, enabling our framework to effectively handle complex and diverse cases. **iii) Task-Specific Structures**. While foundation models offer valuable priors, task-specific components are necessary to tailor their application. Motivated by studies [80, 73, 40], we guide these models toward optimal probability distributions with specialized modules, achieving more accurate and contextually appropriate restorations.

Concretely, we first produce an automatic pipeline to construct paired data, which uses the Boolean modifier in Blender. It offers diverse and large-scale data that are difficult to acquire manually. Second, we propose an innovative framework named **Restore3D**, comprising two key components, *i.e.*, **multi-view image inpainting and reconstruction**. There are several foundational models [52, 31, 69] in these two components that we can leverage prior knowledge to further handle more diverse incomplete objects effectively. However, simply applying foundational models to multi-view images introduces several **challenges**, as shown in Fig. 2, including: *1) View Inconsistency*: Generated results often

differ across views, leading to visual incoherence. *2) Depth Understanding*: Existing models often lack robust depth perception, resulting in failures to recognize occlusions and spatial relationships. *3) Inpainting Position Perception*: Accurately identifying regions requiring inpainting can be difficult, especially for large masks.

To address these issues, we propose a **multi-view** base model combined with a specially designed **mask self-perceiver** module incorporating a **depth-aware mask rectifier**. This module autonomously perceives and reconstructs missing components, preserving the integrity of original broken regions and ensuring consistent results across multiple views. Additionally, by leveraging the preserved and generated masks predicted by the self-perceiver, we can develop an image integration and enhancement pipeline (see Fig. 2 & 4), yielding high-quality and consistent results. To convert high-quality multi-view images into 3D objects, we employ large reconstruction models (LRMs)[23, 56, 69, 29, 63], which offer efficient single- and multi-view object reconstruction capabilities. To overcome the limitation of coarse outputs from these models, we adopt a coarse-to-fine refinement approach. Leveraging recent advances in surface normal prediction models[3, 72], we inject normal priors to progressively enhance geometric quality, and refine texture based on updated geometry by using enhanced images. This ensures that our refined shapes and textures maintain high fidelity, even for complex scenarios.

We conduct extensive experiments on Objaverse [17], GSO [18], and OmniObject3D [67] to validate the quality of inpainting and reconstruction. The results demonstrate that our inpainting method significantly outperforms previous approaches [36, 80, 50], *e.g.*, ↑ 13 in PSNR compared to Nerfiller [62]. By carefully designing a mask self-perceiver, our method can alleviate view inconsistency, understand depth concepts, and capture inpainting regions, achieving consistent structure and texture styles without requiring user-provided masks to indicate inpainting regions. For reconstruction, our approach enhances both geometric and texture quality as shown in Fig. 1, indicating that our proposed framework is capable of producing complete shapes and textures with relatively high fidelity compared to baseline methods [22, 69]. Overall, our contributions are summarized as follows,

- To the best of our knowledge, we are among the first to explore the completion of relatively complex shapes and textures. To support this task, we introduce an automated data synthesis pipeline that generates paired incomplete and complete shapes and textures, providing a rich source of training data named RestoreIt-3D.

- We propose Restore3D, a novel framework to tackle shape and texture completion through a combination of multi-view image inpainting and reconstruction. In multi-view image inpainting, we design a mask self-perceiver with a depth-aware mask rectifier for autonomous perception and reconstruction of missing components, ensuring preservation of original features. Moreover, we introduce an image integration and enhancement pipeline to restore fine details. We refine coarse meshes by using normal priors and enhanced images.

- Comprehensive experiments validate the effectiveness of Restore3D, demonstrating its ability to produce complete and high-quality textured meshes.

## 2   Related Work

**2D Inpainting and Generation models** 2D inpainting methods are designed to complete missing content in an image using a given image and mask. LaMa [54] utilizes fast Fourier convolutions, a large receptive field, and extensive training masks to effectively fill large missing areas, producing plausible inpainting results. Recent advancements in image generation [50, 80] have demonstrated superior performance and can be adapted for inpainting tasks with high-quality outcomes. RePaint [36] modifies the diffusion generation process, allowing it to be used for inpainting. NeRFiller [62] uses grid priors to make the 2D diffusion model produce more consistent multi-view inpainting results. However, these methods require a user-defined mask to specify the regions that need inpainting.

**3D Generation and Completion** Recent 3D generation models [61, 30, 9] showcase promising results. DreamFusion [45] and SJC [59] are first proposed to generate 3D assets from text using the strong 2D text-to-image generation model [50]. As 2D diffusion models easily lead to 3D inconsistency, some works [31, 82, 57, 55, 58, 70] focus on consistent multi-view image diffusion models. MVDream [52] uses 3D self-attention and camera embedding to achieve multi-view text-to-image generation. Considering the time-consuming nature of SDS-based methods, there are some works [20, 34, 29, 33, 56, 65, 35] that use multi-view diffusion models and reconstruction models.

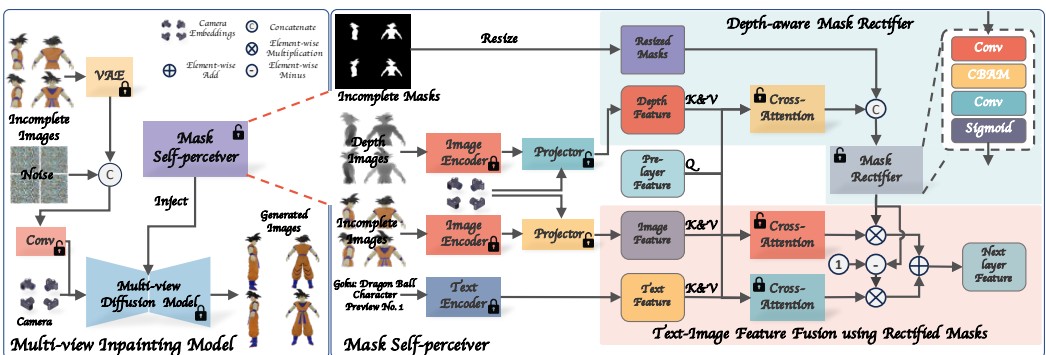

Figure 3: **An overview of multi-view image inpainting.** We carefully design a mask self-perceiver based on a multi-view diffusion model that composes the image and text features with a spatial mask predicted by a depth-aware mask rectifier, therefore the model can automatically perceive the missing part and further generate it meanwhile preserving the original parts.

Another line for 3D generation is that directly train 3D generative models using 3D representations like point cloud [42, 77, 37, 81], meshes [32, 21], neural fields [27, 1, 41, 24, 78, 19, 8]. In addition to 3D generation, recent 3D shape completion works [26, 79, 66, 68, 16, 15, 39, 44, 12, 13] usually use different types of 3D representations and networks to model global and local structures, *e.g.*, point cloud, sdf, GAN, VAE, and diffusion models. However, they all learn models on small-scale datasets, therefore the modeling capacity is limited compared with some 3D generation models trained on large-scale datasets (*e.g.*, Objavese [17]). Moreover, these works do not consider the texture.

**Texture Generation.** Several texture generation works [49, 5, 7] use an iteratively texturing strategy based on the pre-trained depth-to-image diffusion models, yielding high-quality texture. However, these methods tend to error lighting inherited from training data. Paint3D [76] proposes a shape-aware UV Inpainting and a shape-aware UVHD diffusion model to alleviate this situation. There is another line to learn texture. Texturify [53] employs texture maps on the surface of meshes and uses StyleGAN [25] to predict texture. Mesh2Tex [4] incorporates an implicit texture field for texture prediction. These methods are lacking in global information modeling. PointUV [75] first trains a diffusion model specifically for mesh texture generation, and the proposed coarse-to-fine framework allows it to enjoy the efficiency of 2D representation while enhancing 3D consistency. Other approaches like AUV-net [10], LTG [74], and TUVF [11] learn to generate UV-Maps for 3D shapes. However, they typically focus on the texture generation starting from a complete shape.

## 3 Method

### 3.1 Data Preparation

**Motivation.** We browse the datasets of related tasks and find that the existing datasets [6, 17, 67, 18, 14] are not sufficient to handle the shape and texture completion of broken objects, which suggests the need to construct specific broken and complete paired data. However, collecting large-scale paired data in the real world is *time-consuming and labor-intensive*. Thus we propose to *synthesize* broken and complete paired data.

**Data Collection.** We select the recent dataset, G-objaverse [46] that has *more diverse and general objects*, and sample about 83K 3D objects from this dataset.

**Synthesis Pipeiline.** Specifically, we propose an automatic data processing technique using Boolean operations (*i.e.*, Difference and Intersect) of Blender. Additionally, we equip the dataset with text captions using Cap3D [38]. Subsequently, we normalize and merge the prepared 3D data. The use of Boolean operations requires the introduction of another object. Therefore, we use an ico sphere or cube with random size and rotation angle and then randomly place them inside the 3D bounding box of the prepared 3D data to ensure that the objects can be realistically segmented. After that, it is essential to render this processed data in the format of RGB images to facilitate model learning. We execute the rendering at a resolution of 256×256. The camera settings include a randomly chosen elevation between -10° and 30°. Additionally, the azimuth values are uniformly rendered from 0° to

143 $360°$ with a randomly sampled start view, producing a total of 32 images per object. The Fov of the
144 camera is randomly from $35°$ to $45°$ and the distance is always 2.

## 3.2 Multi-view Image Inpainting

**Motivation.** Traditional single-view image inpainting methods [54, 50, 80] rely on the user-provided
masks that indicate the areas to be inpainted. While this approach works well in the context of
single-view images, it presents significant challenges when extended to multi-view contexts as shown
in Fig. 2. *1. View inconsistency.* In a multi-view scenario, the user is required to manually provide a
mask for each of the views (*e.g.*, four views in our case). This also introduces the risk of errors, as
the mask needs to be accurately aligned across different perspectives to maintain 3D consistency. *2.*
*Uncertainty Regarding Inpainting Areas.* These models cannot autonomously perceive the regions
that require inpainting when a large mask is applied. Additionally, they do not incorporate depth
perception, limiting their understanding of occlusion and spatial relationships. To address these
challenges, we propose an innovative approach that enables the model to *ensure view consistency* and
*self-perceive the mask*. Concretely, we design the following two parts.

**Mask Self-perceiver.** We propose a mask self-perceiver module based on a multi-view image
generation model as shown in Fig. 3. It has two projectors that consist of transformer-based blocks
and camera modulation layers, which project the depth and image features $(f_d, f_r)$ extracted from
CLIP [47] to the diffusion feature space. The camera modulation helps the model to discriminate the
feature under different cameras. Then these projected features $(p_d, p_r)$ will be fed to the respective
cross-attention blocks as key and value $(\mathbf{K_d}, \mathbf{K_r}, \mathbf{V_d}, \mathbf{V_r})$. The process can be formulated as follows,

$$p_* = \mathbf{Proj}(f_*, c) = \mathbf{Trans}(\mathbf{Mod}(f_*, c)) \tag{1}$$

$$s_* = \mathbf{Softmax}(\frac{\mathbf{Q}\mathbf{K_*^T}}{\sqrt{d}})\mathbf{V}_* \tag{2}$$

where $f_*$ can be depth or image features, $p_*$ is the projected features of them. Similarly, $s_*$, $\mathbf{K}_*$ and
$\mathbf{V}_*$ are the results of $p_*$ via cross-attention and linear layers. $\mathbf{Q}$ originates from the pre-layer features
in the diffusion model.

**Depth-aware Mask Rectifier.** Since depth effectively captures the incomplete shape while disre-
garding texture information, the rectifier can focus solely on identifying the regions that require
generation and preservation. Moreover, the depth can help the model understand the spatial relation
and occlusion. Specifically, This module leverages depth features obtained after the cross-attention
layer, along with incomplete masks, and inputs them into a mask rectifier. The rectifier then outputs a
mask indicating where needs to be generated *i.e.*, leveraging the text features and where needs to be
preserved *i.e.*, using the image features. The process can be formulated as follows,

$$\mathcal{M}_r = \mathbf{Sigmoid}(\mathbf{Conv}(\mathbf{CBAM}(\mathbf{Conv}[s_d, \mathcal{M}_o]))) \tag{3}$$

$$f_n = (\mathbf{1} - \mathcal{M}_r)s_t + \mathcal{M}_r s_r \tag{4}$$

where $\mathbf{Conv}$ is a convolution layer, and $\mathbf{CBAM}$ is Convolutional Block Attention Module [64]

**Training objectives** Given training samples, including incomplete images $\mathcal{I}$, depth images $\mathcal{D}$,
incomplete masks $\mathcal{M}$, text prompts $\mathcal{P}$ and camera embedding $\mathcal{C}$, the multi-view inpainting loss can
be formulated as follows,

$$\mathcal{L} = \min_\theta \mathbb{E}_{z, \epsilon \sim \mathcal{N}(\mathbf{0}, \mathbf{I}), t}\|\epsilon - \epsilon_\theta(z_t; t, \mathcal{I}, \mathcal{D}, \mathcal{M}, \mathcal{P}, \mathcal{C})\|_2^2. \tag{5}$$

## 3.3 Image Integration and Enhancement

**Motivation.** The input resolution of multi-view model is 256 x 256, which is subsequently encoded
to 32 x 32 using a Variational Autoencoder. As a result, *local details are compressed, leading to a*
*loss of clarity in both the original and generated regions of the image.* This compression often causes
the inpainted part to be unclear, and the reconstructed image may lose fine details that are essential for
achieving high-quality results. Moreover, *high-quality images* will *help* the next *reconstruction* stage
to give accurate and detailed textured meshes. To address these challenges, we propose a pipeline
that enables the model to *restore local details and preserve the original patterns*.

**Enhancement Models.** We explore two types of enhancement models. *Real-ESRGAN* [60] is
effective at preserving the patterns of low-resolution images with minimal misalignment, making
it ideal for recovering the overall structure. *ControlNet-Tile* [80] offers advanced capabilities for
enhancing image details, but will modify the original pattern when a high denoising step is used.

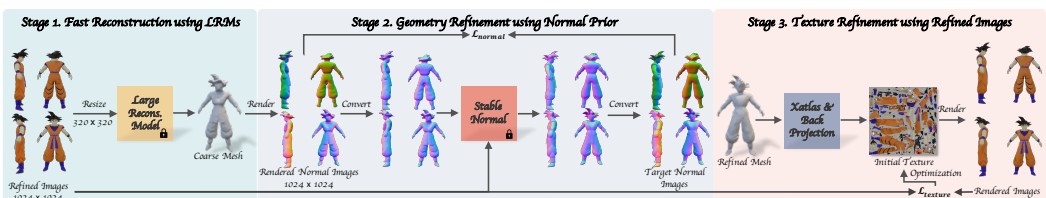

Figure 5: **Geometry and Texture Refinement.** We separately refine the geometry and texture of the coarse results inferred by LRMs [69].

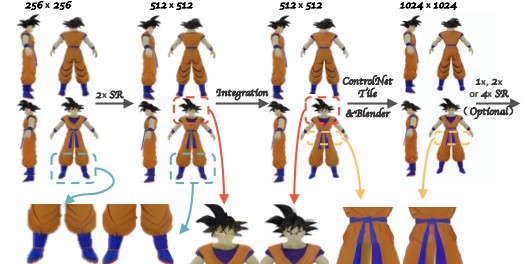

Based on these properties, we design the following enhancement pipeline. *1. Input resolution alignment using Real-ESRGAN.* Before integrating with the original images, we need to align the resolution. Using Real-ESRGAN effectively preserves the overall structure and does not introduce content that is not related to the original style. *2. Integration of generated and original parts using rectified masks.* As depicted in Fig. 4, this procedure infers the preserved and generated masks used to compose the im-

Figure 4: **Image Integration and Enhancement Pipeline using Rectified Masks.**

ages, which preserves the original parts as soon as possible. However, this procedure inevitably leads to some artifacts, *e.g.*, inconsistent color transitions. To address these artifacts, we leverage the mentioned property of ControlNet-Tile to enhance the images. *3. Image harmonizing using ControlNet-Tile with a blending strategy.* Directly using ControlNet-Tile will alter the original pattern and destroy the integration step. Inspired by previous works [2, 36], we incorporate a mask blending technique within the diffusion process. This technique helps maintain the original patterns, eliminates any gaps caused by integration in image space, and enhances the image quality.

## 3.4 Multi-view Image Reconstruction

**Fast Reconstruction using Large Reconstruction Models (LRMs).** Recent advancements in LRMs [23, 56, 69], which leverage sophisticated architectures, large-scale datasets, and extensive model parameters, have demonstrated impressive capabilities in 3D object reconstruction from single or sparse-view images. These models are particularly well-suited for tasks requiring fast mesh reconstruction. However, while LRMs can produce initial reconstructions efficiently, the results are often *coarse and lack the fine details* necessary for high-quality 3D representations. To address this limitation, we adopt a coarse-to-fine schema and refine the shapes and textures of the outputs generated by LRMs, separately, as shown in Fig. 5.

**Geometry Refinement using Normal Prior.** A key component in optimizing shape structure is to obtain high-quality surface normals. Recent surface normal estimation methods [3, 72] have demonstrated the ability to predict relatively accurate normals for in-the-wild monocular images or videos. Therefore, we can employ an *off-the-shelf* normal estimation model to provide normal priors and then use it to optimize the shape structure of 3D objects. Since these models are primarily trained on monocular images or videos, the predicted normals are typically in camera space. Thus we need to convert these normals into world space using camera extrinsic parameters. Specifically, we select StableNorm, a model that accepts coarse rendered normals and RGB images as inputs to predict refined normal outputs. The consistency of the rendered normals contributes to the stability and accuracy of the predicted normals, allowing for more precise geometry refinement.

**Texture Refinement using Enhanced High-quality Images.** Since the current shape differs from the coarse shape, the original texture no longer aligns with the updated geometry. Thus we propose to learn the textures that better match the optimized shape. Concretely, we can use Xatlas to obtain UV coordinates, enabling us to back-project the colors from the inpainted images onto the UV textures. After that, we treat the UV textures as parameters and use the enhanced high-quality images to optimize the texture maps.

**Training Objectives.** We apply a normal loss $\mathcal{L}_{normal}$ based on the rendered normals $\mathcal{I}_n$ and the target normals $\hat{\mathcal{I}}_n$. Additionally, we apply a mask loss $\mathcal{L}_{mask}$ to ensure that the optimization regions

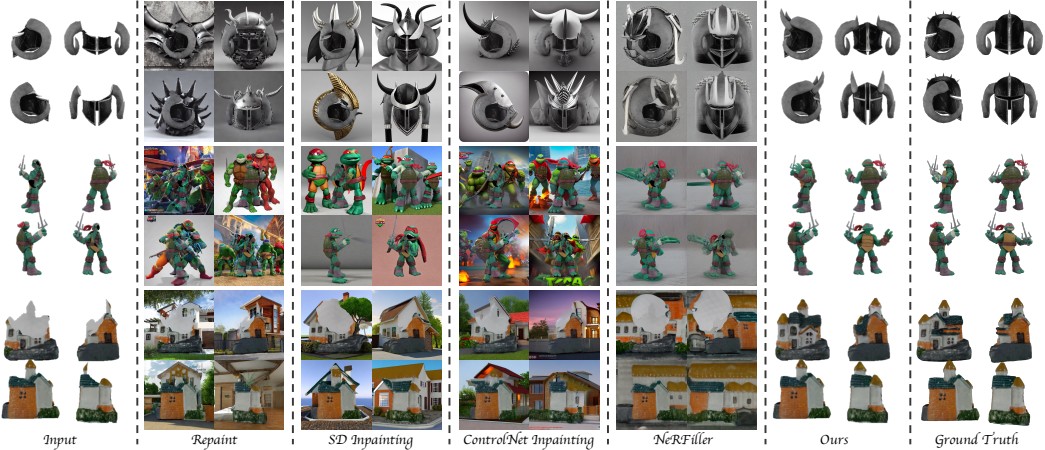

Figure 6: **Visual comparison with inpainting methods.**

Table 1: **Comparison with the previous inpainting and reconstruction methods.** ⋆ means inpainting, while △ means using Depth-Anything [71] to obtain the depth images. Note that we do not apply image integration and enhancement pipelines. IM means InstantMesh [69].



(a) **Inpainting.**

| Method | PSNR ↑ | LPIPS ↓ | FID ↓ | SSIM ↑ |
|---|---|---|---|---|
| Repaint [36] | 10.55 | 0.31 | 69.57 | 0.76 |
| SD ⋆ [50] | 12.58 | 0.22 | 61.15 | 0.83 |
| ControlNet ⋆ [80] | 10.66 | 0.30 | 69.91 | 0.76 |
| NeRFiller [62] | 12.03 | 0.25 | 65.20 | 0.82 |
| Ours △ | 25.29 | 0.07 | 32.05 | 0.95 |
| Ours | **25.50** | **0.06** | **31.82** | **0.95** |

(b) **Reconstruction.**

| Method | PSNR ↑ | LPIPS ↓ | CD ↓ | F-Score ↑ |
|---|---|---|---|---|
| Open-LRM [22] | 16.90 | 0.15 | 0.011 | 0.179 |
| IM [69] | 20.60 | 0.11 | 0.006 | 0.321 |
| Ours | **23.35** | **0.09** | **0.005** | **0.389** |



are correctly aligned. The loss function is defined as follows,

$$\mathcal{L}_{shape} = \mathcal{L}_{normal} + \mathcal{L}_{mask} = \|\mathcal{I}_n - \hat{\mathcal{I}}_n\|_2^2 + \|\mathcal{M} - \hat{\mathcal{M}}\|_2^2. \tag{6}$$

To optimize the texture, we use a RGB loss $\mathcal{L}_{rgb}$ on the rendered images $\mathcal{I}_{rgb}$ and enhanced images $\hat{\mathcal{I}_{rgb}}$. The mask loss $\mathcal{L}_{mask}$ is also applied. Moreover, the SSIM $\mathcal{L}_{ssim}$ loss is introduced to improve the texture quality. The loss functions are defined as follows,

$$\mathcal{L}_{tex} = \mathcal{L}_{rgb} + \mathcal{L}_{mask} + \lambda\mathcal{L}_{ssim} = \|\mathcal{I}_{rgb} - \hat{\mathcal{I}_{rgb}}\|_2^2 + \|\mathcal{M} - \hat{\mathcal{M}}\|_2^2 + \lambda\mathbf{SSIM}(\mathcal{I}, \hat{\mathcal{I}}), \tag{7}$$

where $\lambda$ is a weight parameter.

## 4 Experiments

**Dataset & Metrics.** For model training, we sample approximately 83K data from the G-objaverse [46] dataset and process them using our proposed pipeline. For model testing, we sample approximately 350 data from the GSO [18], Omniobject [67], and Objaverse [17] datasets. **Inpainting.** To assess image quality, We choose Peak Signal-to-Noise Ratio (PSNR), Frechet Inception Distance (FID), Learned Perceptual Image Patch Similarity (LPIPS), and Structural Similarity Index Measure (SSIM). **Reconstruction.** In addition to the metrics mentioned above, we evaluate geometry quality using Chamfer Distance (CD) and F-scores.

### 4.1 Inpainting Results.

**Baselines.** We compare our method with single-view image inpainting, *i.e.*, Repaint[36], Stable-Diffusion [50], Controlnet [80], and a multi-view inpainting method, *i.e.*, Nerfiller [62].

**Qualitative Comparison.** As shown in Fig. 6, the results demonstrate that our model produces plausible and coherent inpainting outcomes. Previous methods require user-provided masks to guide the model in generating missing parts. However, when given a relatively large mask, these methods struggle to capture the inherent structure of the objects, leading to less accurate and coherent inpainting.

Table 2: **Ablation studies for multi-view inpainting and reconstruction.** GR and TR mean geometry and texture refinements.

<table>
<tr><td colspan="4" align="center">(a) Inpainting.</td></tr>
<tr><th>Method</th><th>PSNR ↑</th><th>LPIPS ↓</th><th>SSIM ↑</th></tr>
<tr><td>IF</td><td>22.65</td><td>0.14</td><td>0.90</td></tr>
<tr><td>IF + Conv</td><td>26.53</td><td>0.08</td><td>0.94</td></tr>
<tr><td>IF + Conv + DMR</td><td>29.44</td><td>0.06</td><td>0.95</td></tr>
</table>

<table>
<tr><td colspan="5" align="center">(b) Reconstruction.</td></tr>
<tr><th>Method</th><th>PSNR ↑</th><th>LPIPS ↓</th><th>CD ↓</th><th>F-Score ↑</th></tr>
<tr><td>Baseline</td><td>20.60</td><td>0.11</td><td>0.006</td><td>0.321</td></tr>
<tr><td>GR</td><td>-</td><td>-</td><td>0.005</td><td>0.389</td></tr>
<tr><td>GR + TR</td><td>23.35</td><td>0.09</td><td>0.005</td><td>0.389</td></tr>
</table>

In contrast, our approach does not require predefined inpainting masks. Instead, it autonomously perceives and reconstructs the missing regions, capturing the underlying structure of the object without manual intervention. This capability allows our method to produce high-quality, structurally consistent inpainting results.

**Quantitative Comparison.** As illustrated in Table 1a, we observe the following: **1)** Our approach achieves the best performance in restoring shape and texture. **2)** When applying depth images predicted by Depth-Anything [71], our method yields results comparable to those obtained with ground truth depths. **3)** The compared methods produce noticeably inferior results in terms of inpainting quality.

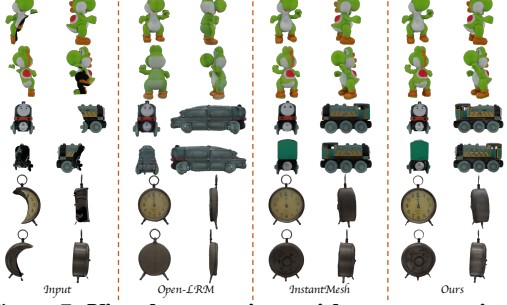

Figure 7: **Visual comparison with reconstruction methods.**

## 4.2 Reconstruction Results.

**Baselines.** We compare our method against both single-view and multi-view LRMs, including LRM [22, 23] and InstantMesh [69]. For single-view baselines, we input the front-view image.

**Quantitative & Qualitative Comparison.** As shown in Table. 1b, our method achieves superior rendered image quality and geometry accuracy, with a substantial improvement over baseline methods. In Fig. 7, it is evident that our approach delivers clearer details and the most accurate geometry among the compared methods. **Training time.** Our approach is highly efficient, requiring 20 seconds per object for geometry and texture refinements.

## 4.3 Ablation Study

**Multi-view Inpainting.** We conduct ablation studies on the proposed multi-view inpainting module in the following components: **1) IF.** Only inputting incomplete images to the cross-attention layers. 2) **Conv.** Concatenating noise and incomplete images to a learnable convolutional layer. 3) **DMR.** Adding the designed Depth-aware Mask Rectifier. As shown in Table 2a, the results improve progressively with each added component, and using all designed components achieves the highest results. In the qualitative comparison shown in Fig. 8b, 1) IF Only: the model captures the general style of the object but lacks an understanding of spatial relationships and structure. 2) IF + Conv: This enables the model to capture spatial positioning and understand object

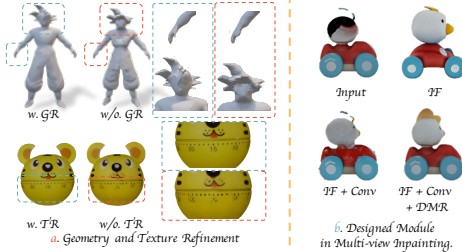

Figure 8: **Visualization of ablation studies.**

structure. However, it is still prone to color inaccuracies, especially in areas like the head (blended with error black color). Additionally, the region that needs to be preserved is changed. 3) IF + Conv + DMR: This allows the model to improve its ability to handle occlusions and spatial relationships, producing the best inpainting quality, with coherent colors and well-preserved spatial structure.

**Reconstruction.** We evaluate the impact of the following components: 1) Geometry Refinement (GR), and 2) Texture Refinement (TR). In Table 2b and Fig. 8a, incorporating GR leads to substantial improvements in geometry quality. TR improves the visual quality of the rendered images.

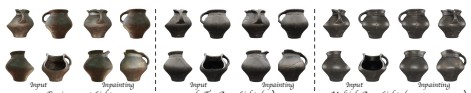

Figure 9: **Different lighting settings.**

Table 3: **Different lighting settings.**

| Method | PSNR ↑ | LPIPS ↓ | SSIM ↑ |
|---|---|---|---|
| Top area light | 25.18 | 0.06 | 0.95 |
| Multiple area lights | 25.50 | 0.06 | 0.95 |
| Environment light | 25.28 | 0.06 | 0.95 |

Table 4: **Occlusion results.**

| Method | PSNR ↑ | LPIPS ↓ | SSIM ↑ |
|---|---|---|---|
| 1-view | 27.16 | 0.06 | 0.95 |
| 4-view | 25.62 | 0.07 | 0.95 |

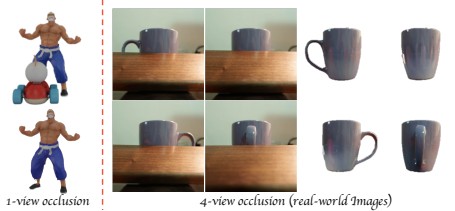

1-view occlusion | 4-view occlusion (real-world Images)

Figure 10: **Visualization of occlusion cases.**

Table 5: **BBD [51] results**

| Method | PSNR ↑ | LPIPS ↓ | SSIM ↑ |
|---|---|---|---|
| SD-inpainting | 12.02 | 0.74 | 0.53 |
| ControlNet | 14.50 | 0.59 | 0.71 |
| NeRFiller | 17.66 | 0.52 | 0.79 |
| Ours | **25.09** | **0.10** | **0.95** |

Input | Inpainting | Input | Inpainting | Input | Inpainting

Figure 11: **Visual results on BBD [51]**.

**Different lights.** We render our test samples with different lights and test our inpainting model on these rendered images. In Table 3 and Fig. 9, the results show our model can achieve promising results under different lighting settings.

# 5 Application

Our Restore3D can be directly used for some applications:

**Object Restoration.** We test our model on the validation set of Breaking Bad Dataset (BBD) [51], as shown in Fig. 11 and Table 5. This dataset is synthesized by a physically based method that simulates the natural destruction process of geometric objects.

**Occluded Object Reconstruction.** We arrange either a single object or four objects to create occluded scenarios with one view and four views, respectively, based on our 350 test samples. As shown in Table 4 and Fig. 10, the results indicate

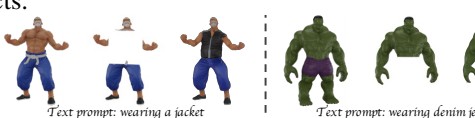

Text prompt: wearing a jacket | Text prompt: wearing denim jeans.

Figure 12: **Text-guided editing results.**

that the one-view occlusion scenario achieves higher performance, as the occluded regions can be inferred more easily from the visible areas. When applying four-view occlusion, our model still demonstrates strong performance. In addition, we present a real-world example in Fig. 10.

**3D Object Editing.** We can position a cube or sphere over the target region for editing and use a Boolean operation to segment the object. This enables us to render the object as an incomplete image. We then process them using our inpainting model with a text prompt for editing. Finally, we apply the reconstruction model. In Fig. 12, our approach successfully handles simple editing scenarios.

# 6 Conclusion

In this paper, we propose a novel framework named Restore3D, consisting of multi-view image inpainting and reconstruction, to simultaneously complete both the shape and texture of broken 3D objects. To facilitate this task, we develop an automated data processing pipeline that collects pair-wise data from a large-scale dataset [17]. In the multi-view image inpainting, we design a mask self-perceiver with a depth-aware mask rectifier. This component autonomously identifies and reconstructs missing regions while preserving the original patterns. To address the low resolution resulting from the base model [52], we implement an image integration and enhancement pipeline, allowing for seamless integration and detail enhancement by learned masks. For the reconstruction stage, we employ an LRM to quickly generate a coarse result, followed by separate geometry refinement using normal priors and texture refinement using enhanced images. Through this designed framework, our model produces coherent completions of broken objects as illustrated in Fig. 1.

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

# A  Technical Appendices and Supplementary Material

## A.1  Preliminary

**Multi-view Diffusion models.** Extending 2D generation models to the multi-view domain has been explored in various works [31, 52]. These extensions often incorporate modifications like adding camera conditions and adjusting the attention mechanisms to enable effective multi-view synthesis. In this paper, we adopt MVDream as our base model. MVDream modifies the spatial attention mechanism in Stable Diffusion [50], allowing the attention to focus on corresponding features across different views.

## A.2  Implementation Details

We train the multi-view inpainting model using four NVIDIA A100 GPUs. We use the Adam optimizer and incorporate classifier-free guidance. The training is conducted with a learning rate of 1e-4 and a batch size of 256. MVDream is utilized as the base model for multi-view inpainting, while InstantMesh is employed as the large reconstruction model. The input consists of 4-view images. For the sampling process, we employ DDIM with 50 steps and a guidance scale of 5.0.

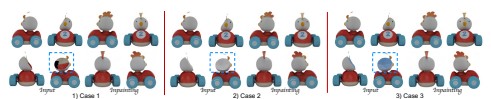

Figure 13: **Different color types.**

Table 6: **Ablation studies of views.**

| Method | PSNR ↑ | LPIPS ↓ | FID ↓ | SSIM ↑ |
|---|---|---|---|---|
| 4-view | 25.50 | 0.06 | 31.82 | 0.95 |
| 6-view | 25.00 | 0.07 | 24.70 | 0.95 |
| 8-view | 25.17 | 0.07 | 20.49 | 0.95 |

## A.3  More Results

**More views.** Although our model is trained on a 4-view setting, our model can be directly used to process inputs with more views. As shown in Table 6, the results show that their performance is comparable to the 4-view setting.

**Different color types on the broken plane.** As shown in Fig. 13, altering the broken plane (blue dotted box) with different colors does not affect our model's ability to complete the broken objects. This further validates that our model effectively distinguishes between regions that need to be preserved and those that require generation.

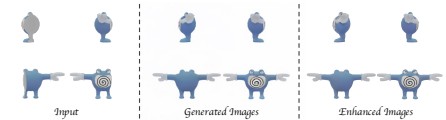

Figure 14: **Visualization of image integration and enhancement**.

**Image Integration and Enhancement** As shown in Fig. 14, we provide some results of this pipeline. The results show that the proposed pipeline restores the original pattern and improves the image quality.

**Inpainting and reconstruction results on full GSO dataset (1030 Objects).** As shown in Table 7 and Table 8, our model achieves the best performance on both inpainting and reconstruction results.

## A.4  Limitations

Our approach builds upon a base model and thus inevitably inherits some of its limitations. For instance, the low resolution of the input restricts the ability to capture very fine details, such as the facial features of characters, even with the application of enhancement techniques. In addition, there is still a lot of room to enrich the quality of geometry and material details in the reconstruction.

Table 7: **Inpainting results on GSO**.

| Method | PSNR ↑ | LPIPS ↓ | FID ↓ | SSIM ↑ |
|---|---|---|---|---|
| SD-inpainting | 13.52 | 0.68 | 67.79 | 0.55 |
| ControlNet | 12.63 | 0.70 | 83.46 | 0.51 |
| NeRFiller | 17.07 | 0.60 | 75.24 | 0.72 |
| Ours | **26.02** | **0.06** | **11.12** | **0.94** |

Table 8: **Reconstruction results on GSO**.

| Method | PSNR ↑ | LPIPS ↓ | CD ↓ | F-Score ↑ |
|---|---|---|---|---|
| Open-LRM | 17.56 | 0.15 | 0.014 | 0.15 |
| IM | 22.15 | 0.11 | **0.002** | 0.36 |
| Ours | **24.74** | **0.08** | **0.002** | **0.43** |

## A.5 Broader Impacts.

Object restoration will help cultural heritage preservation: restoring historical artifacts, sculptures, and architectural elements with accuracy. Negative impact: the ability to create highly accurate replicas can be misused for fraudulent purposes, such as creating counterfeit artifacts, artworks, or products, which can deceive consumers and harm original creators.

