# OpenReview forum: "Restore3D: Breathing Life into Broken Objects with Shape and Texture Restoration"
_NeurIPS.cc/2025/Conference — Submitted to NeurIPS 2025_

### Official Review · Reviewer_SHyz · 2025-06-28

**Clarity:** 3
**Significance:** 2
**Originality:** 3
**Rating:** 4
**Confidence:** 4

**Summary:**

The authors proposed a multi-step method to inpaint the full 3D geometry and texture for partial object from multiview images. They use pretrained models to inpaint the 2D image, lift to 3D, finetune the mesh, and finally optimize the texture.

Contribution:
1.  introduce an automated data synthesis pipeline that generates paired incomplete and complete shapes and texture
2. introduce Restore3D, a pipeline to recover full 3D from partial multiview images
3. validate the effectiveness by various experiments

**Questions:**

Among all weakness in above sections, I think the author should compare the multiview-to-3d steps with current end-to-end baselines, like single RGB to 3D pipelines etc. My main concern is that these end-to-end models may be more straightforward but reach comparable or better result.

**Ethical Concerns:**

["NO or VERY MINOR ethics concerns only"]

**Final Justification:**

Thank the authors for their response to my concerns. I argee with LCYf and TC7X that this paper somehow may needs more real-world experiment. And the pipeline looks somehow complex and redundant to me and may not very easy for other researchers to follow. But as the authors compare with more baselines in rebuttal and claim that they are one of the first teams in this task, I would like to raise my score to borderline accept.

**Limitations:**

yes

**Quality:**

2

**Strengths And Weaknesses:**

Strengths:

1. Good visual result. The model performs better than baselines shown in Fig. 6 and Fig. 7
2. Real-world applications. It is interesting that the model could reconstruct the full objects from occlusion, as shown in Fig. 10, which shows some level of generalizations even using blender to create the training data.

Weakness:

1. The pipeline has 6 different stage, which looks quite complex to me: (1) Inpaint multiview images to complete ones (2) use pretrained  Real-ESRGAN to do super-resolution (3) use ControlNet-Tile to handle the inconsistency during super-resolution (4) use pretrained LRM[69]. model to genrate coarse mesh and render normal (5) use pretrained StableNormal to refine the previous mesh (6) optimize the texture with the input image.

I feel these pretrained model are not tightly associated. For example, the input image after super-resolution (1024*1024) are resized back to low resolution (320*320) and feed to LRM, making the previous super-resolution a bit of unnecessray. And why not train a model to inpaint model and do super-resolution at the same time? Since super-resolution introduces many artifacts, the authors need to use another model to rectify that.

And I think currently there are many end to end models that can handle step 4-6 with good performance. For example, what's the performance compared with other single-view to 3D pipeline? like [Direct3D](https://nju-3dv.github.io/projects/Direct3D/) and [TRELLIS](https://microsoft.github.io/TRELLIS/)? Though they have different number of inputs, they could generate high quality textured 3d objects too.

2.  Does the pipeline require known camera during evalutation? The source of "Camera Embeddings" in figure 3 is not clear. And line 223-225 mentions that the model "need to convert these normals into world space using camera extrinsic parameters". If camera matrix are provided by user, then it will make the application . If the camera matrix are predicted, then this step may be sensitive to the inaccruate camera pose.

3. The authors creating incomplete dataset using random size ico sphere or cube (line 138), which may causes the cross-section to be too smooth and regular, and may not able to fully reflect the missing appearance of real objects.

4. (Minor Weakness) Writing issue.
In figure 1, what does "Complete3D" refer to? it is "Restore3D" (the proposed method)?

---

> ### Author Rebuttal · Authors · 2025-07-29
>
> We sincerely appreciate **your recognition of our methodology and your positive feedback on the details, experiments, and analysis**. Thank you very much for your valuable and encouraging comments!
>
> We will address all the concerns point by point.
>
> -----------------------------------------------------------------------------------------------------------------------------------------------
>
> **W1 & Q1**.
> 1. The framework primarily consists of **multiview inpainting and reconstruction**, which makes it **relatively straightforward** from this perspective. We will clarify it in the revision.
> 2. **Previous super-resolution is not unnecessary.** The **high-resolution images** refined by super-resolution are used in the **geometry and texture refinement stage** to improve the details of the coarse meshes inferred by LRM. As shown in **Figure 8(a) and Table 2(b)** of our paper, the high-resolution images play an important role in coarse mesh refinement.
>
> 3. **Why not train a model to inpaint the model and do super-resolution at the same time?**
>
>     - Inpainting and super-resolution have inherently **different objectives and feature demands**. Inpainting focuses on **semantic hole-filling and structural reconstruction**, requiring strong **contextual modeling** and high-level understanding of scene semantics. In contrast, super-resolution emphasizes **pixel-level detail enhancement and high-frequency texture recovery**. These intrinsic differences make it challenging to design a unified architecture that performs both tasks effectively.
>     - Training a single model to handle both inpainting and super-resolution significantly **increases computational demands**. In **multi-view settings**, these **costs** scale even more **severely** compared to **single-view** configurations. Furthermore, as resolution increases, the computational cost grows exponentially. Consequently, the resolution of the output will be constrained to a relatively low level and still require the addition of a secondary model for further high-resolution enhancement.
>     - **Decoupling inpainting and super-resolution** into separate, modular components enables **greater flexibility and maintainability**. Each module can be independently developed, fine-tuned, or replaced with more advanced techniques as they become available.
>
>
> 3. **Comparison with direct3d and trellis.** We input the front-view image inpainted by our model into direct3d and trellis.
> Since Direct3D does not release its texture generation code, we report only geometry results for that method.  The results show that our model outperforms both methods.
> These baselines are designed for single-view reconstruction, which naturally limits their performance compared to our approach that leverages multi-view inputs. We will add this comparison in the revision.
> | Model    |  CD  |F-score| PSNR | LPIPS |
> |----------|------|-------|------|-------|
> | Direct3D | 0.006 |  0.297    | -      | -     |
> | Trellis  | **0.005** | 0.335 | 21.78 | 0.12 |
> | Ours     | **0.005** | **0.389** | **23.35** | **0.09** |
>
>
>
> -----------------------------------------------------------------------------------------------------------------------------------------------
>
>
> **W2. Camera.** Yes, the pipeline requires a known camera during evaluation. The source of "Camera Embeddings" in Figure 3 is the known 4-view camera parameters.  We will clarify it in the revision.
>
> -----------------------------------------------------------------------------------------------------------------------------------------------
>
> **W3. The cross-section is too smooth and regular, and may not be able to fully reflect the missing appearance of real objects.**
>
> We conduct experiments on the following datasets. As shown in **Figure 11 of our paper** and **Figure 1 of our supplementary material**, both of these datasets include **non-regular break patterns**, further validating the **robustness** of our approach.
> 1. Synthetically fractured objects. As shown in **Table 5 and Figure 11** of our paper, we test our model on Breaking Bad Dataset[1]. This dataset is synthesized by a physically based method that simulates the natural destruction process of geometric objects.
>
> 2. Real-world objects. In **Figure 1 and Table 1** of our **supplementary material**, we include experimental results regarding Fantastic Breaks[2]. This dataset contains 150 paired real-world broken objects.
>
> These experiments demonstrate the **generalization** ability of our model to both **real-world scenarios and unseen physically simulated cases**, validating its **robustness and practical applicability**.
>
> Thank you for highlighting this issue. We will reorganize these contents to enhance their visibility and ensure better clarity.
>
> #### [1] Breaking bad: a dataset for geometric fracture and reassembly. NIPS 2022.
> #### [2] A dataset of paired 3d scans of real-world broken objects and their complete counterparts. CVPR 2023.
>
> -----------------------------------------------------------------------------------------------------------------------------------------------
>
> **W4. Typos.** "Complete3D" is "Restore3D"; we will correct this in the future version of the paper.

---

> > ### Comment · Reviewer_SHyz · 2025-08-03
> >
> > Thanks for the detailed explainations from the authors. I agree that the seperate multi-stage pipeline makes it easier to train or replace each module with limited time or computation resources, but I still think the complexity of this pipeline make the researcher hard to follow this work in the future. And does the model output fixed view only (like fixed 4 views) or aribitraty view (if camera pose is known?). And it's happy to see that the author compare with the direct3d and trellis. It would be great to see the quantative result and more comparable result in the revised version.

---

> > ### Author Response · Authors · 2025-08-04
> >
> > We would like to thank the reviewer for the valuable feedback.
> >
> > 1. We fully understand the concerns raised about the overall complexity of the pipeline.
> > - Currently, this task remains inherently challenging, as it requires not only preserving the original structural patterns but also accurately restoring the broken parts in a way that aligns with those patterns. Given the limitations of existing methods, we introduce refinement modules to enhance the quality of inpainting and reconstruction. While this adds complexity, it is a necessary trade-off to achieve high-fidelity results. We believe that future advances may eventually enable an end-to-end solution that eliminates the need for such modular refinements.
> >
> > - We will **release the full code** of our model and the data processing pipeline, with **clear documentation and usage instructions**.
> >
> > 2. As shown in Table 6 of the Appendix (located at the end of the paper), we present experiments using 6-view and 8-view configurations. These results demonstrate that our model, although trained on 4-view settings, can generalize well to a higher number of input views.
> >
> > 3. We appreciate the reviewer’s suggestion and will include quantitative comparisons with TRELLIS and Direct3D in the revised version to provide a more comprehensive evaluation.

---

> > > ### Comment · Reviewer_SHyz · 2025-08-05
> > >
> > > Thanks for the detailed clarifications!

---

> > > > ### Author Response · Authors · 2025-08-07
> > > >
> > > > Thank you again for your insightful feedback. We plan to revise our paper accordingly based on your comments.
> > > >
> > > > Would you agree that our rebuttal has addressed the concerns you raised? If there’s anything unclear or unresolved, we’d be happy to clarify or further revise.
> > > >
> > > > We sincerely appreciate your time and thoughtful input during the discussion phase.

---

### Official Review · Reviewer_LCYf · 2025-06-30

**Clarity:** 3
**Significance:** 2
**Originality:** 2
**Rating:** 3
**Confidence:** 4

**Summary:**

In this paper the authors have proposed a method, Restore3D, for shape and texture completion of incomplete 3D objects, using consistent multi-view inpainting and Large Reconstruction Models (LRMs). At the core of the method is a mask prediction module which predicts multi-view consistent masks from a single image, crucial for consistent multi-view inpainting. A LRM uses the completed multi-view images to produce 3D reconstruction of objects. Finally, a quality enhancement pipeline, renders the recovered object into normal images and employs normal image enhancement and texture enhancement modules to improve the shape and texture quality of the 3D reconstruction. The method achieves improved performance in comparisons with recent SOTA for the task of 3D object completion.

**Questions:**

- Regarding the reconstruction results in section 4.2, what is the input of the other methods? Do you use your inpainting to first complete the image and pass the same input to all methods?

- Line 277. Training time should be Inference time?

**Ethical Concerns:**

["NO or VERY MINOR ethics concerns only"]

**Final Justification:**

I thank the authors for their response to my comments. It still seems to me that alignment between the motivation of the paper and the experiments section is moderate, leading to an incoherent paper overall. As also mentioned by reviewer TC7X, the paper lacks experiments on real cases, which is the main motivation. Therefore, I will keep my rating as Borderline Reject.

**Limitations:**

yes

**Paper Formatting Concerns:**

No formatting concerns

**Quality:**

2

**Strengths And Weaknesses:**

Strengths:
- The multi-view mask prediction with the Mask Self-Perceiver module is a novel ideal as far as I am aware, for multi-view consistent inpainting. The implementation is interesting and according to the provided results is seems to be working.

- Multi-view inpainting demonstrates improved results compared to SOTA.

- The authors provided an ablation study which helps to understand the individual effect of the employed design choices.

Weaknesses:
- The motivation of the paper, as well as the problem statement is not aligned with the conducted work and experiments. The paper is presented as a method for 3D reconstruction of fractured objects and states the importance of restoring incomplete or damaged objects. However the work is a 3D object completion method and the conducted experiments mainly focus on synthetic occlusions rather than real.

- The evaluation part of the paper lacks of  experiments on real or synthetically fractured objects, again diverging the method from the initial motivation.

- The method produces higher quality 3D reconstructions than LRM and InstantMesh, however coarse-to-fine quality improvement is not a novel idea as has been done in a similar way in [*1]. Also, recent related methods such as [*1, 2, 3] about enhancing an initial 3D reconstruction are not discussed or compared with in the paper.

[*1] Wu K, Liu F, Cai Z, Yan R, Wang H, Hu Y, Duan Y, Ma K. Unique3d: High-quality and efficient 3d mesh generation from a single image. In The Thirty-eighth Annual Conference on Neural Information Processing Systems 2024 Jan 1.

[*2] Luo Y, Zhou S, Lan Y, Pan X, Loy CC. 3DEnhancer: Consistent Multi-View Diffusion for 3D Enhancement. In Proceedings of the Computer Vision and Pattern Recognition Conference 2025 (pp. 16430-16440).

[*3] Edelstein Y, Patashnik O, Cohen-Bar D, Zelnik-Manor L. Sharp-It: A Multi-view to Multi-view Diffusion Model for 3D Synthesis and Manipulation. In Proceedings of the Computer Vision and Pattern Recognition Conference 2025 (pp. 21458-21468).

---

> ### Author Rebuttal · Authors · 2025-07-29
>
> We sincerely appreciate **your recognition of the novelty of our designed method and your positive feedback on the details, experiments, and analysis**. Thank you very much for your valuable and encouraging comments!
>
> We will address all the concerns point by point.
>
> -----------------------------------------------------------------------------------------------------------------------------------------------
>
> **W1. Inconsistency of motivation and experiments.**
> 1. Our primary contribution is that we are **among the first to explore the restoration or completion of the shape and texture in broken 3D objects**.
> Based on this, we propose a **well-structured and effective pipeline** tailored for this task. We conduct **comprehensive experiments** to assess the **performance and generalization capability** of our model across diverse scenarios.
>     - As shown in **Table 1 and Figures 6** of our paper, we evaluate our model on datasets generated using the **same synthetic method employed during training**.
>     - As shown in **Table 5 and Figure 11** of our paper, we further test our model on the Breaking Bad Dataset[1] synthesized by a **physically based** method, **different from the synthesis method used in our training**.
>     - As shown in **Table1 and Figure 1** of the **supplementary material**, we also  evaluate our model on **real-world** broken objects.
>
>     These experiments demonstrate the **generalization** ability of our model to both **real-world scenarios and unseen physically simulated cases**, validating its **robustness and practical applicability**.
>
> 2. **Occlusion handling** is not a primary focus of our work. It is a **byproduct** of our model's design, as a minor contribution, evaluated in Table 4 and Figure 10.
>
> Thank you for highlighting this issue. We will **reorganize these contents to enhance their visibility and ensure better clarity**.
>
> -----------------------------------------------------------------------------------------------------------------------------------------------
>
> **W2. Experiments on real or synthetically fractured objects**
> 1. Synthetically fractured objects. As shown in **Table 5 and Figure 11** of our paper, we test our model on Breaking Bad Dataset[1]. This dataset is synthesized by a **physically based** method that **simulates the natural destruction process** of geometric objects.
>
> 2. Real-world objects. In **Figure 1 and Table 1** of our **supplementary material**, we include experimental results regarding Fantastic Breaks[2]. This dataset contains 150 paired **real-world broken objects**.
>
> Both of these results show the **generalization** ability of our model. Thank you for highlighting this issue. We will **reorganize these contents to enhance their visibility and ensure better clarity**.
>
> #### [1] Breaking bad: a dataset for geometric fracture and reassembly. NIPS 2022.
> #### [2] A dataset of paired 3d scans of real-world broken objects and their complete counterparts. CVPR 2023.
>
> -----------------------------------------------------------------------------------------------------------------------------------------------
>
> **W3. Comparison with coarse-to-fine methods.**
> 1. Coarse-to-fine is not our main contribution. Our primary contribution is that we are **among the first to explore the restoration or completion of the shape and texture in broken 3D objects**. Based on this, we propose a **well-structured and effective pipeline** tailored for this task. Central to our method is the **Mask Self-Perceiver** module, which generates masks that guide a consistent super-resolution process. This results in **high-quality image outputs**, enabling **robust multi-view consistent inpainting**.
> 2. Coarse-to-fine plays a crucial role, as shown in Figures 4 & 8 of our paper. By progressively refining the reconstruction, it significantly **enhances performance** and leads to more **faithful and realistic restoration results**. Thus, it remains a necessary and impactful design choice within our overall pipeline.
>
>
> - **3DEnhancer**. 3DEnhancer is a multi-view image enhancement method. We compare it with our image integration and enhancement pipeline, and our results demonstrate superior performance. It is worth noting that 3DEnhancer is limited to a resolution of 512 pixels, whereas our method supports resolutions of 1K and can be scaled up to 2K and even 4K. Moreover, 3D Enhancer can not preserve the texture of the original broken part if it does not use our predicted mask.
> | model | PSNR |LPIPS|SSIM|
> |-|-|-|-|
> | 3D Enhancer| 26.37 | 0.07 | 0.96 |
> | Ours | **26.97**| **0.05**| **0.97**|
>
> - **Unique3D**. We replace the single-image to multi-view model in Unique3D with our inpainting model. Experimental results indicate that our method achieves better reconstruction performance.
> | Model    |  CD  |F-score| PSNR | LPIPS |
> |----------|------|-------|------|-------|
> | Unique3D  | **0.005** | 0.306 | 22.00 | 0.14 |
> | Ours     | **0.005** | **0.389** | **23.35** | **0.09** |
>
> - **Sharp-It**. The author does not release their code, so we are unable to conduct a direct comparison.  We will cite their work and discuss it in the Related Work section.
>
> Thank you for highlighting this issue. We will **include these experiments and discuss them in the related work**.
>
> -----------------------------------------------------------------------------------------------------------------------------------------------
>
> **Q1**: Yes. The input of the other methods is the inpainted and enhanced images generated by our model.
>
> -----------------------------------------------------------------------------------------------------------------------------------------------
>
> **Q2**: In our opinion, for per-object reconstruction, the training time refers to the optimization of both geometry and texture, while the inference time corresponds to the rendering of the final mesh.

---

> > ### Author Response · Authors · 2025-08-04
> >
> > We sincerely thank you for your time and thoughtful feedback. We have carefully considered and addressed the concerns raised in our rebuttal.
> > As the discussion period concludes in three days, please feel free to let us know if any further clarification is needed. We sincerely appreciate your engagement.

---

> > > ### Author Response · Authors · 2025-08-08
> > >
> > > We sincerely thank you for your time and thoughtful feedback. We have carefully considered and addressed the concerns raised in our rebuttal. As the discussion period concludes in **24 hours**, please feel free to let us know if any further clarification is needed. We sincerely appreciate your engagement.

---

> ### Comment · Area_Chair_CUyj · 2025-08-07
> **Please engage to the discussions!**
>
> Dear Reviewer,
>
> I would like to invite you to the discussions with the authors. At least, please carefully read the others' reviews and authors' responses, and mention if the rebuttals addressed the concerns or not.
>
> To facilitate discussions, the Author-Reviewer discussion phase has been extended by 48h till Aug 8, 11:59 pm AoE; but to have enough time to exchange opinions, please respond as quickly as possible.
>
> Thanks,
>
> Your AC

---

### Official Review · Reviewer_F5uK · 2025-06-30

**Clarity:** 2
**Significance:** 3
**Originality:** 2
**Rating:** 4
**Confidence:** 4

**Summary:**

Collaborative restoration of complex object shape and texture.
this work designs a mask self-perceiver module with a depth-aware mask corrector to autonomously perceive and reconstruct missing parts, ensuring that the original features are preserved, and develops an image fusion enhancement pipeline to restore fine details.
At the same time, a large number of comparative experiments and ablation studies were conducted to prove that the above method is reasonable and efficient.

**Questions:**

- For the inpainting experiment, other multiview inpainting methods are needed for comparison, such as using controlnet's mvpaint instead of the single-view version of SD
- We need to add more experiments. For example, in the image infilling stage, use a single view to infill the first image, and use the SOTA multiview method to generate the rest. For example, if we get multi-view input at the very beginning, then directly use LRM to reconstruct, and then use the mask of the front view to project to other views, what will be the result (using the above method to infill)?
- The training data are all synthetic data, but in actual use, it is very likely that they are not synthetic data. Are there any experimental results for this data gap? Please provide some analysis of failure cases.
- As far as I know, the current high-resolution multi-view reconstruction has achieved a resolution of 1k to 2k (that is, 2k to 4k multi-view). How is the difference in details between the multi-stage coarse-to-fine scheme used in this work and these high-resolution schemes?

**Ethical Concerns:**

["NO or VERY MINOR ethics concerns only"]

**Final Justification:**

I appreciate the authors' reply. I feel that all my questions have been properly addressed, and on the whole, this paper makes a solid contribution to the relevant field.

I vote Borderline accept

**Limitations:**

- From the pipeline, the input set in this work is multi-view input, which is a rare case. In this case, it should be convenient to perform a geometric reconstruction first and then inpaint, but the author did not do this.
- The coarse-to-fine solution only increases the resolution to 512, which cannot reach the 1k or 2k resolution level of SOTA.
- This work mentions that this method is needed in areas such as cultural relics restoration, but there is no relevant content in the experimental results and applications

**Paper Formatting Concerns:**

Do not cover the image with words in the figure.
Do not use cursive fonts in images

**Quality:**

3

**Strengths And Weaknesses:**

Strengths:
- The first batch of work exploring the collaborative restoration of complex object shapes and textures
- No need for manual mask annotation, with automatic mask generation mechanism, to achieve higher pipeline automation

Weaknesses:
- The experimental part lacks some comparisons with other related works. For example, in the inpaint experiment, there should be comparisons with other multiview methods, not just comparisons based on SD.
- The fonts and diagrams in the charts are poor, and many contents need to be compared with the article repeatedly to understand them.
- A key comparison was not made: after processing with the single-image mask inpaint solution, the SOTA multiview generation solution was used for generation and reconstruction.

---

> ### Author Rebuttal · Authors · 2025-07-30
>
> We sincerely appreciate **your recognition of our methodology and the proposed task**. Thank you very much for your valuable and encouraging comments!
>
> Our **core contribution** is that we are among the first to explore the restoration or completion of the shape and texture in broken 3D objects. Based on this, we develop a data synthesis pipeline to construct training pairs and propose a well-structured and effective pipeline tailored for this task. Extensive experiments show the performance and generalization capability of our model across unseen physically simulated cases and real-world cases.
>
> We will address all the concerns point by point.
>
> -----------------------------------------------------------------------------------------------------------------------------------------------
>
> **W1 & Q1. Comparison with multiview inpainting methods**
>
> 1. As shown in **Table 1 and Figure 6 of our paper**, we include comparative results with the **multi-view inpainting method** Nerfiller [1], not limited to single-view settings.
> 2. Additionally, we present new comparisons with Instant3dit [2], a multi-view inpainting method. Unlike our approach, Instant3dit requires a **manually defined mask** to specify the inpainting region. When we use a simple inverted mask of the broken object as input, Instant3DIt performs noticeably worse than our method. When leveraging the inpainting mask predicted by our model, Instant3dit sees significant performance improvements. Even so, it continues to underperform relative to our approach.
> | model | PSNR |LPIPS|SSIM|
> |-|-|-|-|
> | Instant3edit | 19.40 | 0.10 | 0.94 |
> | Instant3edit (w/ our predicted mask) | 22.37 | 0.07 | 0.95 |
> | Ours | **25.50**| **0.06**| **0.95**|
>     #### [1] Nerfiller: Completing scenes via generative 3d inpainting. CVPR, 2024.
>     #### [2] Instant3dit: Multiview Inpainting for Fast Editing of 3D Objects. CVPR 2025.
>     **We will clarify it and include these experiments in the revision**.
>
> -----------------------------------------------------------------------------------------------------------------------------------------------
>
> **W2. Fonts and diagrams.**
> We apologize for the poor quality of the figures and fonts in the charts.
> - We will update the fonts.
> - We will adjust the text that currently covers the images.
> - Regarding the figures, could you please indicate which parts are unclear and share any suggestions you may have for improvement? We understand that the multiview inpainting pipeline may seem complex, and we plan to reorganize it to enhance clarity. For example, we will first illustrate a block in Unet that has three different types of cross-attention connected by our Mask Rectifier, and then we will provide details of each cross-attention layer.
>
> -----------------------------------------------------------------------------------------------------------------------------------------------
>
> **W3 & Q2 example1. Single-view inpainting + image-to-multiview.**
>
> "Single-view inpainting + image-to-multiview generation" is inherently less accurate than multiview inpainting approaches, as the synthesized views produced by image-to-multiview techniques often misalign with the ground-truth multiview images.
>
> We first apply SD-inpainting and pix2gestalt [2] to the front-view image, followed by an SOTA image-to-multiview method, MV-Adapter[1]. However, as illustrated in Figure 2 of our paper, single-view inpainting methods often encounter **specific limitations**, such as Depth Understanding and Inpainting Position Perception. Consequently, the multiview images generated by MV-Adapter tend to exhibit degraded quality.
>
> **We will clarify it and include these experiments in the revision**.
>
> #### [1] MV-Adapter: Multi-view Consistent Image Generation Made Easy. ICCV 2025
> #### [2] pix2gestalt: Amodal Segmentation by Synthesizing Wholes. CVPR 2024
> | model | PSNR |LPIPS|SSIM|
> |-|-|-|-|
> | SD-inpainting + MV-adapter | 13.27 | 0.27 | 0.80 |
> | pix2gestalt + MV-adapter | 16.43 | 0.21 | 0.86 |
> | Ours | **25.50**| **0.06**| **0.95**|
>
> -----------------------------------------------------------------------------------------------------------------------------------------------
>
> **Q2 example 2.**
> 1. LRM is unable to accurately capture the shape of broken objects.
> 2. The front-view mask does not indicate the regions that require inpainting. As a result, the outputs are essentially the same as directly inputting the multiview images into the inpainting model.
> 3. Please refer to the example in Figure 6 of our paper (Row 3 Input). Ideally, the goal is to inpaint the gray region. However, the approach you mentioned fails to infer the correct inpainting mask corresponding to this gray area.
>
> If we have misunderstood your idea, please feel free to correct us. We will proceed with the experiments accordingly.
>
> -----------------------------------------------------------------------------------------------------------------------------------------------
>
> **Q3. Experiments on real and synthetically fractured objects.**
>
> 1. Synthetically fractured objects. As shown in **Table 5 and Figure 11** of our paper, we test our model on Breaking Bad Dataset[1]. This dataset is synthesized by a physically based method that simulates the natural destruction process of geometric objects.
>
> 2. Real-world objects. In **Figure 1 and Table 1** of our **supplementary material**, we include experimental results regarding Fantastic Breaks[2]. This dataset contains 150 paired real-world broken objects.
>
>     These experiments demonstrate the **generalization** ability of our model to both **real-world scenarios and unseen physically simulated cases**, validating its **robustness and practical applicability**.
>     **We will incorporate these experiments into the main paper to enhance their visibility and ensure better clarity**.
>     #### [1] Breaking bad: a dataset for geometric fracture and reassembly. NIPS 2022.
>     #### [2] A dataset of paired 3d scans of real-world broken objects and their complete counterparts. CVPR 2023.
>
> 3. Failure cases. The main failure cases are concentrated in regions that originally contained rich details but have been severely damaged, e.g., the face or head of dolls. As a result, the inpainting process struggles to fully reconstruct these areas, often leading to suboptimal or undesirable outcomes. **We will add this discussion to the limitations**.
>
> -----------------------------------------------------------------------------------------------------------------------------------------------
>
> **Limitation 1**.
> For the real-world dataset Fantastic Breaks, the objects are already pre-scanned, so geometric reconstruction is not required in this case. We can easily get the 4-view images from the scanned objects. In practice, we can also take a 4-view photo using a phone, as shown in Figure 10 of our paper. It is also sufficient to achieve good reconstruction results.
>
> -----------------------------------------------------------------------------------------------------------------------------------------------
>
> **Q4 & Limitations 2. High-resolution**.
> 1. Our image integration and enhancement pipeline achieves a resolution of **1K** in all experiments, as demonstrated in **Figures 4 and 5** of our paper, rather than being **limited to 512**.
> 2. In most multiview settings, e.g., 3D enhancer, Unique3D, Hunyuan3D, the resolutions of the multiview model are limited to 512 or lower. After that, they will use single-view super-resolution methods (e.g., Stable-Diffusion Upscale 4x, Controlnet Tile, or RealESRGAN) to enhance resolution.
> 3. Our pipeline adopts Real-ESRGAN and Controlnet tile as the backbone, which is capable of scaling outputs up to 2K or 4K resolution. However, in our case, we observe no substantial improvement in visual detail at 2K compared to 1K. Furthermore, optimizing textures at 2K introduces significantly higher GPU memory consumption and training time, which we find impractical given the marginal visual gains relative to the 1K resolution.
>
> -----------------------------------------------------------------------------------------------------------------------------------------------
>
> **Limitation 3. Cultural relics restoration.**
>
> Our training and testing dataset includes objects with shapes and textures **similar to those of cultural relics**, such as vases and statues. This research may contribute to the restoration of cultural relics.
>
>
> **Thank you for pointing out the limitations. We will include them in our list.**

---

> > ### Author Response · Authors · 2025-08-08
> >
> > We sincerely thank you for your time and thoughtful feedback. We have carefully considered and addressed the concerns raised in our rebuttal. As the discussion period concludes in **24 hours**, please feel free to let us know if any further clarification is needed. We sincerely appreciate your engagement.

---

> ### Author Response · Authors · 2025-08-04
>
> We sincerely thank you for your time and thoughtful feedback. We have carefully considered and addressed the concerns raised in our rebuttal.
> As the discussion period concludes in three days, please feel free to let us know if any further clarification is needed. We sincerely appreciate your engagement.

---

> ### Comment · Area_Chair_CUyj · 2025-08-07
> **Please engage to the discussions!**
>
> Dear Reviewer,
>
> I would like to invite you to the discussions with the authors. At least, please carefully read the others' reviews and authors' responses, and mention if the rebuttals addressed the concerns or not.
>
> To facilitate discussions, the Author-Reviewer discussion phase has been extended by 48h till Aug 8, 11:59 pm AoE; but to have enough time to exchange opinions, please respond as quickly as possible.
>
> Thanks,
>
> Your AC

---

### Official Review · Reviewer_TC7X · 2025-07-03

**Clarity:** 3
**Significance:** 3
**Originality:** 3
**Rating:** 5
**Confidence:** 4

**Summary:**

The paper proposes a system for reconstructing 3D objects from multi-view images of partial (aka broken) objects. I find this to be an interesting problem due to its applicability in heritage restoration and occluded object reconstruction which is highly relevant for domains like robotic vision. Moreover, the system is capable of generating texture in addition to geometry. The paper proposes to achieve this by leveraging several vision foundation models and combining them with custom made modules like Mask Self-perceiver and Depth-aware Mask Rectifier to achieve state of the art performance. Extensive comparisons with state of the art methods demonstrate the superiority of Restore3D. Several ablations are also included to help understand component contributions.

**Questions:**

See weakness section.

**Ethical Concerns:**

["NO or VERY MINOR ethics concerns only"]

**Final Justification:**

I thank the authors for their response. I believe all my concerns have been clarified and overall I think this paper makes a solid contribution to the community. I would encourage other reviewers who have given borderline rejects to kindly note that the paper is perhaps one of the first ones to solve a unique problem and as a community we should be more accommodating of that. The authors have given substantial experimental results to back up their claims.

I vote to accept this paper.

**Limitations:**

yes

**Quality:**

3

**Strengths And Weaknesses:**

Strengths:

1. The problem of reconstructing complete 3D assets from multiple images of broken objects is a novel problem with several applications. Restore3D generates visually superior geometry and texture results compared to prior state of the art. Table 1 and Figure 6 corroborate paper's claims of being SOTA for multi-view image inpainting.

2. Clever use of Foundation Models to achieve reconstruction of geometry and texture. The paper proposes Mask self-perceiver and depth-aware mask rectifier for effective combination of priors from foundation models to perform image Inpainting. Their contributions are mentioned in Table 2 and Figure 8.

3. The method uses standard tricks in reconstruction like iterative refinement to get better results. Moreover the lack of need for manually defining the masks further helps with the usability of the method.


Weakness:

1. Lack of real-world examples both for training as well as inference. I am afraid that the paper hasn't explored the real world use case a bit even though this is one of the core motivations for their problem. It would have been interesting to see model trained on synthetic data deployed on real world multi-view images of broken objects.

2. Material synthesis is not physically based. Given that the model is trained on synthetic dataset, I wonder what is the difficulty in generating albedo, specularity, etc.

3. Needing to run multiple VFMs, means heavy GPU requirements. It would be good to have a discussion on the efficiency viz baseline methods.

---

> ### Author Rebuttal · Authors · 2025-07-29
>
> We sincerely appreciate **your recognition of the novelty of our proposed problem and your positive feedback on the details, experiments, and analysis**. Thank you very much for your valuable and encouraging comments!
>
>
> We will address all the concerns point by point.
>
> -----------------------------------------------------------------------------------------------------------------------------------------------
>
> **W1. Real-world examples.**
>
> As shown in **Figure 1 and Table 1 of our supplementary material**, we present experimental results on the Fantastic Breaks dataset [1], which contains 150 pairs of **real-world broken objects**. For instance, although the fracture surfaces in this dataset are often **irregular and unsmooth**, our model is still able to perform **promising inpainting**. These results demonstrate the **generalization** ability of our model to real-world scenarios, despite being trained solely on synthetic data.
>
> Thank you for highlighting the experiments involving real-world broken objects. We will **incorporate these experiments into the main paper** to enhance their **visibility** and ensure better **clarity**.
>
> #### [1] A dataset of paired 3d scans of real-world broken objects and their complete counterparts. CVPR 2023.
>
> -----------------------------------------------------------------------------------------------------------------------------------------------
>
> **W2. Physical-based material synthesis**.
> 1. Although our model does not currently predict material attributes, our **primary contribution** lies in introducing a new task, **simultaneously restoring the shape and texture of broken objects**, and **designing an effective pipeline**. Hence, the objective of this task is to generate a complete, textured 3D mesh from partial observations, which is not an easy task. We also conduct extensive experiments to validate the effectiveness of our proposed pipeline.
>
> 2. At present, material attribute inpainting is significantly more **difficult** than simply predicting color. **Designing a model capable of simultaneously inpainting material properties is inherently complex**. Such a model must capture intricate interrelationships between different material attributes while preserving the structural integrity of each individual material. This **increases the overall model complexity**, requires **more computational resources**, and **complicates the training process**.
>
> 3. Recent studies [1,2] have shown the potential of visual foundation models in **estimating material properties**. We believe these models can be effectively leveraged in a **post-processing** stage to enhance the realism of generated textured meshes.
>
> 4. In future work, we plan to design a physics-based material inpainting model, building upon both these foundation models and the insights gained from our Restore3D pipeline.
>
> Thank you for highlighting this issue. We will **include this discussion in the limitations and future work section**.
>
> #### [1] Material Anything: Generating Materials for Any 3D Object via Diffusion. CVPR 2025
>
> #### [2] Neural LightRig: Unlocking Accurate Object Normal and Material Estimation with Multi-Light Diffusion. CVPR 2025
>
> -----------------------------------------------------------------------------------------------------------------------------------------------
>
> **W3. Efficiency.**
>
> During inference, our model does not require substantial GPU resources. A **single NVIDIA RTX 3090 GPU (24GB)** is sufficient for running the entire pipeline efficiently.
>
> We report the per-object inference time for each component of our method as follows:
> 1. For inpainting, our base model (MVDream) is fine-tuned from Stable Diffusion (SD), and shares the same network architecture as SD. Compared to the standard SD inpainting time of 2 seconds, the additional inference time in our method is primarily due to the introduction of our proposed Mask Self-Perceiver and Cross-Attention mechanism.
> 2. For reconstruction, our approach requires approximately 20 seconds per object, which is higher than the LRM baseline (6s). However, this additional time is justified by the significant performance improvements achieved.
>
> In summary, our model is **computationally efficient**, runs on modest GPU memory, and delivers **high-quality results**.
>
> Thank you for mentioning the efficiency. We will add this discussion to the revision.
> | Methods | Time |
> |-|-|
> | Inpainting | 5s |
> | Integration and enhanment | 13s|
> | Coarse mesh reconstruction|6s|
> | Geometry and texture refinement| 20s|
> | total | 44s |

---

> ### Comment · Area_Chair_CUyj · 2025-08-07
> **Please engage to the discussions!**
>
> Dear Reviewer,
>
> I would like to invite you to the discussions with the authors. At least, please carefully read the others' reviews and authors' responses, and mention if the rebuttals addressed the concerns or not.
>
> To facilitate discussions, the Author-Reviewer discussion phase has been extended by 48h till Aug 8, 11:59 pm AoE; but to have enough time to exchange opinions, please respond as quickly as possible.
>
> Thanks,
>
> Your AC

---

### Comment · Area_Chair_CUyj · 2025-08-01
**Author-Reviewer Discussion Period (July 31 - Aug 6)**

The author rebuttals are now posted.

To reviewers:
Please carefully read the *all* reviews and author responses, and engage in an open exchange with the authors.
Please post the response to the authors as soon as possible, so that we can have enough time for back-and-forth discussion with the authors.

---

> ### Comment · Area_Chair_CUyj · 2025-08-05
> **Discussion Period Ends Soon (Aug 6)!**
>
> Dear reviewers,
> Thanks so much for reviewing the paper. The discussion period ends soon. To ensure enough time to discuss this with the authors, please actively engage in the discussions with them if you have not done so.

---

### Note · Authors · 2025-08-15

We thank the AC and all reviewers. We (i) consolidate cross-reviewer strengths and (ii) clarify points that may have caused misunderstandings, alongside concrete actions reflected in the rebuttal and planned revision.

**Strengths**
- **Problem novelty & scope**. Restoring **both geometry and texture** of broken objects from multiview inputs is timely and impactful.
- **Practicality.** **Automatic mask and image generation** and a modular pipeline leveraging foundation models were highlighted as effective and usable.
- **Quality & evidence.** Reviewers acknowledged **visually superior** reconstructions, strong multiview inpainting, occluded-object recovery, and informative ablations.

**Clarifications & actions**
- Generalization beyond synthetic. The paper and supplement already include **unseen physically simulated and real-world** cases; We will **promote key content to the main text** for visibility.
- Multiview inpainting. We compare against Nerfiller (**multiview**) and **add Instant3DIt (CVPR’25)**; moreover, **single-view inpainting + projection** consistently underperforms true multiview.
- Reconstruction. Under the same inputs, our pipeline **outperforms** coarse-to-fine (Unique3D, 3D enhancer) and single-stage methods (Direct3D, TRELLIS); we will **consolidate these results** in the revision.
- Resolution–efficiency trade-off. We report that ~**1K** strikes the best quality/compute balance; moving to **2K** yields marginal texture gains but significant memory/time costs—numbers will be surfaced clearly.
- View-count generalization & reproducibility. Although trained on 4 views, the model **generalizes to 6/8**; we will **release code** and streamline stage dependencies for clarity. We also **tighten wording** on cultural-heritage relevance to avoid over-claim.

We understand the workload and appreciate all efforts. **Not all reviewers could join the discussion**, which limited opportunities to close the loop on several points. We posted detailed clarifications and additional experiments during the window and will integrate them into the revision so that the AC can weigh these materials alongside the initial comments.

With these clarifications and revisions, we believe the paper’s contribution is clear: a practical, automatic, and empirically strong pipeline that unifies multiview inpainting and 3D reconstruction to **restore both shape and texture** of broken objects, with demonstrated generalization and comprehensive comparisons.

---

### Decision · Program_Chairs · 2025-09-17

**Decision:**

Reject

**Comment:**

This paper proposes a method for reconstructing 3D objects from multi-view images of partial (i.e., broken) objects. The method is the collaborative reconstruction of (multi-view) texture and 3D shape. Experiments show better performance compared to existing multi-view inpainting and 3D reconstruction methods.

The reviewers valued the novelty of the task, i.e., multi-view inpainting + 3D reconstruction. However, they are concerned about the novelty and complexity of the proposed method, as well as the lack of real-world experiments. During the reviewer-author discussions, the authors added several important experiments. Finally, the rating is mixed (A, BA, BR, BA).

As the authors mentioned, we acknowledge that the method achieves efficient and high-quality results. Meanwhile, although some reviewers state their acceptance recommendation is based on the task's novelty, e.g., "claim that they are one of the first teams in this task" (Reviewer SHyz), the concept of combining inpainting and 3D reconstruction has been studied. A recent example is [a] (cited as [1] in the main paper but not explicitly compared in the experiment), but not limited to it. (NeRFiller, which is introduced as a multi-view inpainting method, is also considered a one.)

[a] Anciukevičius, et al. "Renderdiffusion: Image diffusion for 3D reconstruction, inpainting and generation." CVPR 2023.

Therefore, we should consider discounting the core novelty of the paper. Although the proposed method should have practical usefulness and high fidelity, we would suggest that authors reorganize their contribution and experiments to strengthen the paper for potential submission to a future venue.